# Red for “Stop”: “Traffic-Light” Nutrition Labels Decrease Unhealthy Food Choices by Increasing Activity and Connectivity in the Frontal Lobe

**DOI:** 10.3390/nu12010128

**Published:** 2020-01-02

**Authors:** Xuemeng Zhang, Yong Liu, Yan Gu, Shaorui Wang, Hong Chen

**Affiliations:** 1Key Laboratory of Cognition and Personality (Ministry of Education), Southwest University, Chongqing 400715, China; zxuemengzfb@163.com (X.Z.); psyliuy@email.swu.edu.cn (Y.L.); shaoruiwang@yeah.net (S.W.); 2School of Psychology, Southwest University, Chongqing 400715, China; guyan323@163.com

**Keywords:** food decision making, nutritional labels, traffic light labels, fMRI

## Abstract

Food labels comprise a national health-intervention policy that informs consumers of food-product nutritional value. Previous evidence has indicated that, compared to a purely numeric guideline-daily-amount label, a traffic-light-inspired, color-coded label more effectively conveys the nutritional level and increases the selection of healthier products. Therefore, we used functional magnetic resonance imaging to assess the mechanism whereby traffic-light and guideline-daily-amount labels influence food-related decision-making. Forty-four female dieters (age, mean = 20.0, standard deviation = 1.45 years) were recruited to participate in a food-choice task; healthy or unhealthy food options were presented with color-coded traffic-light or purely numeric guideline-daily-amount labels, and the participants were asked to state their preference. We found that, compared with the guideline-daily-amount label, a salient, red traffic-light label potentially reduced unhealthy food-related decision-making and activated the superior medial frontal gyrus and the supplementary motor area, which are implicated in the execution of responses and motor inhibition. For the same stimulus contrast, we also found increased activation in the anterior cingulate cortex, which is associated with salient information monitoring. Finally, we found stronger functional connectivity between the anterior cingulate cortex and inhibitory regions (inferior and middle frontal gyri) under red-traffic-light than under guideline-daily-amount label conditions. These results suggest that traffic-light-inspired labels may be a more effective means of public-policy intervention than are numeric labels conveying guideline daily amounts.

## 1. Introduction

Food information labels are critical in informing individuals of the ingredients and nutritional value of their food. Effective food labeling systems help alleviate the prevalence of obesity and support healthy eating decisions [1,2,3,4]. Systematic reviews have shown that nutrition labels are perceived as highly credible and are used to guide food selections [5,6], especially when the nutritional value of a food is ambiguous [7]. Another review concluded that including more salient information and ensuring a deeper understanding of the information on nutrition labels are more likely to promote healthy decision-making [8]. Therefore, in addition to the guideline-daily-amount (GDA) labels that are typically used, the traffic-light (TL) nutrition label has been widely suggested as a more salient adjunct label.

The GDA label displays both the amount per serving and the corresponding percentage of the recommended daily intake for several nutrients. This differs from the TL label, which employs a color-coded system to depict the amount of nutrients per 100 g. Nutritional information such as sugar, fats, and sodium content are identified as red (unhealthy or high levels), yellow (somewhat healthy or medium levels), or green (healthy or low levels). Several meta-analyses have reported that color-coded TL labels, compared to purely numeric GDA labels, are more effective in conveying nutritional levels and increase the selection of healthier products [6,9]. Empirical research has consistently concluded that TL labels are an effective approach in promoting awareness of the health costs of products [4,10], encouraging healthier choices [4,11], increasing the value computation of healthy foods [12], increasing the consumption and sale of healthy items [10,13], and decreasing the consumption and sale of unhealthy items [13,14]. Additionally, a modelled comparison showed that TL labels are a cost-effective method for preventing obesity [15].

Studies have begun to examine various mechanisms to determine why TL labels are more effective in the promotion of health considerations. Eye-tracking studies have revealed that the use of colors in nutrient-specific labels may attract more attention to nutritional information, which may lead to an alternative choice [16,17]. More specifically, one study used drift-diffusion modeling to investigate how color-coded labels alter the process of decision-making [18]. They found that color-coded labels led a significantly increased rate of preference formation (drift rate) toward healthier options without altering the starting point, which was followed by the observation of healthier food selections. Moreover, salient labels increased health sensitivity and decreased the importance of taste, indicating that the integration of health and taste attributes during the choice process is sensitive to how information is displayed [18].

An alternate explanation of the effect of nutrition labels may be the different cognitive neural processes triggered by each type of label; however, most functional magnetic resonance imaging (fMRI) studies have focused on the effect of the caloric and taste information provided by labels on neural responses. Conflicting findings have been observed when foods are described as low in calories on the nutrition label. One study reported that “low calorie” labels did not enhance the activity of the reward-related ventromedial prefrontal cortex (vmPFC), while another found that the participants made fast approach movements to earn the food item, accompanied by an increased activation in the sensorimotor cortex [19,20]. It is considered that these differences are due to individual food taste preferences. In another study, researchers found that by integrating food taste and health attributes on labels, the food taste profiles increased the appreciation and appeal of the food, while the health features increased the proportion of healthy decisions. In the condition where the food was exclusively described with health attribute labels, the activity in the amygdala was stronger. Additionally, the amygdala’s functional connectivity with the dorsolateral prefrontal cortex, a region related to inhibitory control, was stronger [21], indicating that health information provided by labels can affect the brain and guide healthy food decisions. However, these studies lacked discussion of the types of nutrition information labels. To the best of our knowledge, only two studies have compared TL and GDA labels and how they influence neural activity during the evaluation process to determine the value and healthiness of food products [12,22]. Red TL labels activated portions of the left inferior frontal gyrus (IFG)/dorsolateral prefrontal cortex, a region implicated in self-control, and enhanced its coupling with the vmPFC, which is associated with valuation. When presented with green labels, the posterior cingulate cortex showed increased coupling with the valuation system in the vmPFC [12]. These results suggest that using salient nutrition labels triggers neurobiological processes that resemble the self-regulation of eating. However, another study observed no differences in cerebral activation between TL and GDA labels; compared to GDA labels, TL labels did not affect the strategy employed [22]. Therefore, the effect of the two types of nutrition labels on neural activity needs to be analyzed further.

Previous work has left several questions unanswered. First, it is unclear whether TL and GDA labels trigger different cognitive neural activities in dieters processing nutritional information because most food label studies have focused on the general population. Dieting is a commonly used method for weight control [23], and dieters typically display a more urgent desire to reduce their intake of foods high in calories and fat and often rely more heavily on food labels than does the general population. Therefore, it is necessary to investigate the effectiveness of TL and GDA labels in participants who are chronic dieters.

Second, it remains unclear how TL and GDA labels influence neural activity during the food choice process. Decision-making relies on a network of brain regions based in the frontal lobe [24,25,26,27,28]. In particular, the anterior cingulate cortex (ACC) is prominent within a network of regions that monitor the dynamic outcome of actions and guide appropriate decision-making [25,26,27,28,29,30,31]. TL labels are considered more salient than GDA labels, which may direct attention toward the nutritional aspects of the food, help evaluate choice outcomes, and alter ACC activity during the decision-making processes. Red TL labels for unhealthy foods help convey possible negative health consequences and possibly strengthen inhibitory responses to these foods. Inhibitory frontal lobe regions, including the superior medial frontal gyrus, the supplementary motor area (SMA), and the IFG, aid in response or motor inhibition [32,33,34,35,36,37,38]. In the rejection of unhealthier options, response inhibition and self-control may be indispensable; the activity of these inhibitory frontal regions may be stronger under the red TL label condition than under the corresponding GDA label condition. In response to an unexpected event, discontinuing an action requires the outcome of the action be monitored and inhibited. Therefore, red TL labels appear to enhance the functional connectivity between brain regions associated with information monitoring and inhibitory control. In contrast, green TL labels help signify more nutritious foods and their health benefits. Previous studies have used fMRI methods to show that labels indicating low-fat and healthy foods increase activity in the vmPFC and amygdala, which are involved in reward expectation and emotional processing [12,21]. Accordingly, it is considered that green TL labels increase the activity in these regions.

Therefore, the present study aimed to use fMRI techniques to determine how TL and GDA labels influence food decision-making and cognitive neural activity, specifically in chronic dieters. We hypothesized that, among dieters, red TL labels would elicit increased activation compared to GDA labels in brain regions associated with salient information monitoring and response inhibition, whereas green TL labels would elicit relatively increased activation in reward regions.

## 2. Materials and Methods

### 2.1. Participants

The present study included 44 right-handed, neurologically and psychologically healthy female dieters who were recruited from Southwest University in Chong Qing, China. Because restricted diet is common among women and male dieters are relatively rare [39], to avoid imbalance in the number of recruited men and women, only women were selected to participate in this study. Moreover, to avoid the effects of age, our study sample was limited to college students (age, mean (M) = 20.0, standard deviation (SD) = 1.45). Based on previous studies, the Restrained Eating subscale of the Dutch Eating Behavior Questionnaire (DEBQ) was used to screen chronic dieters. The inclusion criterion for participation was an average score greater than 3 (the midpoint on a 5-point scale) on the Restrained Eating subscale [40,41,42]. An exclusion was a reported history of, or currently suffering from, an eating disorder. Obese recruits (body mass index (BMI) > 30 kg/m^2^) were excluded because of potential neuro-anatomical variations as a function of BMI. We also excluded potential participants who had followed a medically prescribed diet within 6 months prior to enrollment, as well as those with preferences for vegetarian food because this factor could affect dietary choices. Forty-eight women were finally recruited; however, the data from four participants were excluded because of the presence of large head movements during fMRI scanning.

### 2.2. Procedure

The experiment was approved by the Southwest University Human Ethics Committee and performed in accordance with the guidelines of the Declaration of Helsinki. Written consent was obtained from each participant upon arrival at the testing location. Participants were asked to refrain from eating or drinking (except water) for at least 2 h before arriving at the laboratory to control the hunger level [42]. Next, participants were familiarized with the fMRI paradigm through practice on a computer. Subsequently, they were asked to complete a visual analog scale (VAS) to assess individual and subjective differences in hunger, eating desire, and happiness. Following this, participants underwent four fMRI runs of a nutrition label decision-making task. Each run included healthy and unhealthy foods with a TL or GDA label and lasted 398 s. Each run contained 24 trials, including four types of conditions. The experimental trials were randomly arranged in advance to ensure that the experimental conditions were not repeated multiple times in succession, so that different experimental conditions would appear intermittently. In the interval between runs, participants briefly closed their eyes to rest. The entire procedure lasted approximately 30 min per participant. Finally, participants were required to describe their impression of the study to ensure that they had not been able to identify the true purpose of the experiment. Participants verbally reported that they had referenced nutrition labels in making their food choices but had not known the meaning of the color of the label.

### 2.3. Materials and Nutrition Label Decision-Making Task

The nutrition label decision-making task involved the presentation of two types of nutrition labels (TL and GDA) and two types of foods (healthy and unhealthy). We employed food pictures adopted from previous studies [41,43] comprising 24 unhealthy food choices (high-calorie and high-fat; e.g., fried foods, cream cakes, Chinese braised pork) and 24 healthy food choices (low-calorie and low-fat; e.g., vegetables and fruits). To avoid forcing participants to select foods that they disliked, only those food pictures that had scored 4 or higher on the tastiness scale (range: 1, not tasty at all to 9, very tasty) in the previous studies were included [41,43]. The degree of tastiness was not significantly different for unhealthy versus healthy foods (*M* = 6.50, SD = 0.98 versus *M* = 6.04, SD = 0.86; *t* (46) = 1.73, *p* = 0.09). The labels described the nutritional content of energy, fat, carbohydrates, and sodium, which are typically seen on labels of Chinese products. Nutritional information was extracted from the website www.boohee.com, which is a well-known food database in China. In accordance with the methods utilized by Enax et al. [12], the TL nutrition labels of unhealthy and healthy foods were primarily red and green, respectively. The GDA labels of the unhealthy and healthy foods were not saliently color-coded as were the TL labels. The two types of labels are shown in Figure 1A.

Participants were instructed to observe the processed food items, were provided with their nutritional value, and then asked to indicate whether they wanted the food. There were four runs consisting of a total of 96 trials, with each run including four conditions. Each picture was shown twice but in different runs: one presentation with the GDA label and the other with the TL label. Food pictures were not repeated in a run. This was performed to ensure that food choices were influenced by different labels, not by different foods. Trials in each run were presented in a predetermined manner, illustrated in Figure 1B. Each food and nutrition label were presented for 5 s, followed by a blank screen for 3–5 s. Participants were then allowed 3 s to indicate whether they wanted to eat the food. Each trial was separated by a fixation cross presented for 3–5 s. Half of the participants were randomly assigned to press the “left” key if they wanted to eat the food or the “right” key if they did not; the other half were asked to press the opposite keys.

### 2.4. Measures

#### 2.4.1. Restrained Eating Subscale of the DEBQ

The Restrained Eating subscale of the DEBQ consists of 10 items (e.g., “When you put on weight, do you eat less than you usually do?”) that instructs participants to rate each item on a 5-point scale from 1 (*never*) to 5 (*very often*). Higher scores on this subscale are indicative of higher degrees of restraint in eating behavior. This scale has a reported internal consistency α of 0.95 [44]; however, in the present study, its internal consistency was α = 0.78.

#### 2.4.2. Visual Analog Scale

A VAS was used to assess individual differences in hunger, eating desire, and happiness. Participants were asked: “How hungry are you currently?,” “How strong is your eating desire currently?,” and “How happy are you currently?”. The questions were answered by selecting a point on a straight line representing a VAS score ranging from 0% to 100%.

### 2.5. fMRI Data Acquisition

Images were obtained using a Siemens Allegra 3 T head-only MRI scanner. Functional data were acquired with T2 *-weighted gradient echo planar imaging (EPI) sequences. For each participant, 199 brain volumes were collected during each functional run. EPI sequences used the following parameters: repetition time (TR) = 2000 ms, echo time (TE) = 30 ms, flip angle = 90°, field of view = 220 × 220 mm^2^, matrix size = 64 × 64, in-plane resolution = 3 × 3 mm^2^, 32 interleaved 3-mm-thick slices, and inter-slice skip = 0.99 mm. To provide detailed anatomical images aligned to functional scans, high-resolution T1-weighted images were obtained using a magnetization-prepared rapid gradient echo (TR = 2530 ms, TE = 3.39 ms, inversion time = 1100 ms, flip angle = 7°, resolution matrix = 256 × 256, slices = 176, slice thickness = 1.33 mm, and voxel size = 1.3 × 1 × 1.3 mm^3^).

### 2.6. Behavioral Data Analysis

For compatibility with analysis of variance models, which require continuous rather than dichotomous variables for parametric assumptions to be met, the decision-making results were converted to percentage choice scores by calculating the number of times a food was chosen divided by the total number of trials. A 2 × 2 repeated measures analysis of variance was then conducted for food decision-making, with label (TL versus GDA) and food (unhealthy versus healthy) as factors and BMI, hunger and eating desire as covariates.

### 2.7. fMRI Data Analysis

#### 2.7.1. fMRI Data Analysis Processing

All preprocessing steps were carried out using the toolbox for data processing and analysis for brain imaging [45] based on the SPM8 software package (Wellcome Trust Centre for Neuroimaging, London, UK). Slice timing was used to correct slice order, the data were realigned to adjust for any motion, and the first five volumes of the functional runs were discarded to achieve magnetically steady-state images. Images were then normalized to the Montreal Neurological Institute space with a 3 × 3 × 3 mm^3^ voxel size. The normalized data were spatially smoothed with a Gaussian kernel; the full width at half maximum was 6 × 6 × 6 mm. After preprocessing, participants whose head movement exceeded 2.5 mm on any of the six head-motion parameters were excluded.

#### 2.7.2. General Linear Model (GLM) Analyses

After preprocessing, SPM8 was used to analyze the imaging data. Two contrasts were specified in the first-level analysis to determine individual differences in activation: unhealthy foods, TL versus GDA labels; and healthy foods, TL versus GDA labels. In addition, six ongoing motion parameters obtained during realignment were included as no interest variables. All these eight regressors were convolved with the canonical hemodynamic response function and restricted to a whole-brain, gray-matter mask. A GLM then generated statistical parametric maps. The resulting single-participant contrast images were then entered into a second-level, random-effects group analysis for each of the corresponding contrasts. The original voxel-wise FDR considers that voxels are independent of each other. However, voxels are not independent of each other after fMRI data smoothing, which causes the FDR obtained using this voxel-wise FDR correction to be very high for an activated brain region. The topological false discovery rate (FDR) is based on the gaussian random field theory, which considers that voxels are not independent. Voxels beyond the T threshold constitute a cluster, and then FDR is used to correct these clusters, so that the independent units (used to be the total number of voxels, but now is the number of clusters) are greatly reduced [46]. Therefore, the statistical threshold was set at a whole-brain corrected value with a topological FDR of *p* < 0.001 to visualize the main effects.

#### 2.7.3. Psychophysiological Interaction (PPI) Analyses

We performed PPI analyses to investigate how activity within brain regions is modulated as a function of the label condition. Because ACC was the core brain region of food decision-making, it was selected as the seed area to perform the PPI analyses. The PPI models applied at the single-participant level included the following main regressors. The first regressor consisted of the time series of activity in the seed brain area (the ACC) identified in a separate analysis, as described below. The time series was extracted in each participant by drawing a 6-mm sphere around the peak voxels from the second-level analysis, which was then used to guide the isolation of each participant’s peak voxels within that sphere. The second regressor consisted of a task-related contrast (unhealthy foods—TL label versus unhealthy foods—GDA label), and the third regressor consisted of the interaction between the first and second regressors. The contrast maps for the PPI analyses were obtained from the same first-level model as the contrast maps for the other analyses as described in Section 2.5, Section 2.7.1, and Section 2.7.2. The TL vs. GDA contrast for healthy foods was not used for the PPI analyses because we did not find any significant whole-brain clusters, including in the ACC’s response to healthy foods with TL and GDA labels, and thus could not extract the signal value of the ACC (described in Section 3). Therefore, PPI analysis was only conducted for unhealthy foods with TL and GDA labels. The statistical threshold was initially set at a whole-brain corrected value with a topological FDR of *p* < 0.001 to visualize main effects. However, as no brain regions were found at the 0.001 level, we further set it to *p* < 0.005 to explore the results.

## 3. Results

### 3.1. Effect of Label Condition on Decision-Making

We observed a significant main effect of label, *F*(1,43) = 4.580, *p* = 0.039, *η*^2^ = 0.108, in that foods with GDA labels were more often chosen (*M* = 0.66, SD = 0.29) than were those with TL labels (*M* = 0.59, SD = 0.35). The main effect of food was significant, *F*(1,43) = 8.819, *p* = 0.005, *η*^2^ = 0.188, because healthy foods were more frequently chosen (*M* = 0.85, SD = 0.14) than were unhealthy foods (*M* = 0.40, SD = 0.29). We also found a significant two-way interaction between label and food, *F*(1,43) = 5.067, *p* = 0.030, *η*^2^ = 0.118. More unhealthy foods were selected in the GDA label condition (*M* = 0.46, SD = 0.28) than in the TL label condition (*M* = 0.33, SD = 0.30), *F*(1,43) = 21.639, *p* < 0.001, *η*^2^ = 0.363; however, a significant difference was not found for healthy foods with GDA labeling (*M* = 0.85, SD = 0.14) vs. TL labeling (*M* = 0.86, SD = 0.13), *F*(1,43) = 1.221, *p* = 0.276, *η*^2^ = 0.031 (Figure 2). These results suggest that TL labels on unhealthy foods are more effective in reducing unhealthy dietary choices.

### 3.2. Effect of Label Condition on Regional Brain Activity

Whole brain analysis showed increased activation in the ACC, superior medial frontal, SMA, and the fusiform, lingual, and middle occipital gyri in response to unhealthy foods with TL labels vs. GDA labels (topological FDR *p* < 0.001; Table 1, Figure 3). However, the contrast between healthy foods with TL vs. GDA labels did not elicit whole-brain significant clusters (topological FDR *p* < 0.001). These results suggest that the TL labels were possibly superior in enhancing the activity of inhibitory frontal regions only when co-presented with images of unhealthy foods.

### 3.3. Effect of Label Condition on the Functional Connectivity of the ACC

Given that the ACC is prominent within a network of regions that monitor the dynamic outcome of actions and guide decision-making, this region may show label-dependent functional connectivity with other inhibitory regions. We tested this hypothesis using a functional connectivity analysis with the ACC as the seed area. Therefore, a PPI analysis was performed to test for a stronger correlation between the ACC and other inhibitory frontal regions with unhealthy foods and TL labels vs. GDA labels. This analysis revealed stronger positive coupling between the ACC and inferior frontal and middle frontal gyri in unhealthy foods with TL labels condition than in the unhealthy foods with GDA labels condition (Table 1, Figure 4).

## 4. Discussion

To the best of our knowledge, the present study is the first to use fMRI techniques to investigate and compare the effectiveness of information-based and more visually salient nutrition labels among dieters during food decision-making processes. Consistent with our hypothesis and compared to a purely information-based GDA label, the more salient red TL labels for unhealthy foods reduced the frequency with which dieters selected unhealthy foods and increased the activity in the ACC, superior medial frontal gyrus, and SMA. Additionally, the ACC exhibited a stronger functional connectivity with the IFG and MFG when presented with a red label. However, we failed to find evidence for the effectiveness of green nutrition labels in promoting healthy dietary choices, and for healthy foods, the neural processes associated with reward did not differ when participants saw a purely information-based label vs. a more salient nutrition label.

Similar to the way a red traffic light delivers a signal to stop the car, the foods with red TL labels may signal participants to avoid them. Previous studies have found that conveying information on food (such as a high calorie content) to indicate that it is unhealthy, or highlighting labels with traffic-signal colors, directly reduced unhealthy eating behaviors [4,11,47]. Our results are similar to these findings, in that TL labels were found to be a potential approach to reducing unhealthy choices among dieters.

More importantly, we found increased activation in the ACC, superior medial frontal gyrus, and SMA in response to unhealthy foods with red TL labels than with GDA labels. The stages-of-change model of Prochaska and DiClemente emphasizes that individuals weigh the advantages and disadvantages during the decision-making process, considering the rewards and risks of each option [48]. This implies that the process of decision-making involves activity in the ACC because this is a region that monitors the dynamic outcome of decision-making [25,26,27,28,29,30,31]. Moreover, in a study by Enax et al. the task required participants to perform a value computation of food products. This task mainly activated the vmPFC, a brain region consistently associated with value computations across task modalities [12]. In the present study, we did not find activity within this region, likely because our tasks, and therefore the involved cognitive processes, greatly differed from those of Enax et al. However, similarly to these researchers, we found that cognitive neural processes associated with inhibitory control are important in conditions involving red TL labels. In addition to the increased activation of the IFG discovered by Enax et al. other inhibitory regions including the superior medial frontal gyrus and SMA, were more active in the red TL label condition than in the respective GDA-label condition. This provides potential evidence that red TL labels for unhealthy foods convey a negative health message, thereby allowing individuals to refrain from selecting such foods, despite employing value computation in the decision-making process. Furthermore, Enax et al. found that the vmPFC, which is related to value computations, exhibited stronger functional connectivity to the IFG in the red TL label condition than in the respective GDA label condition [12]. Accordingly, we found that the ACC was associated with monitoring the dynamic outcome of the decision-making process and also exhibited stronger functional connectivity with the IFG and MFG during the red TL label condition. Our results showed that the red TL labels for unhealthy foods enhanced the corresponding monitoring of decision-making and the activation response in behavioral inhibition regions during the food selection process. Additionally, the red TL labels promoted functional connectivity between the corresponding food value computations and activity in inhibitory brain regions, possibly contributing to the reduction of unhealthy decisions. Taken together, the results constitute further evidence for the potential effectiveness of red TL labels and how this type of labeling can prevent unhealthy eating.

However, the green TL nutrition labels did not positively promote healthy food decisions or increase the activity in reward-expectation-related brain regions compared to the respective GDA labels, possibly because of factors related to participant selection and the experimental material. First, the selected healthy foods had a high degree of appeal to avoid forcing participants to choose foods they did not like. Second, our participants were long-term dieters, and thus most of the healthy foods were palatable to them and compatible with their goal of weight loss [49]. Therefore, our data may be showing a ceiling effect caused by the selection of palatable, healthy foods, regardless of the label condition. As a result, a nutrition labeling effect was not seen during the process of healthy food decision-making, and we did not observe any differences in neural activity or behavior under the two labeling conditions. Moreover, highly palatable but unhealthy foods are not conducive to a dieter’s weight control; therefore, such foods were selected to a lesser degree than healthy foods. Especially, dieters reduced their choice of unhealthy foods under the red TL label than under the GDA label condition, suggesting that the red TL nutrition label could effectively activate cognitive resources to prevent the participant from selecting the food. While green TL labels failed to strengthen the selection of palatable, healthy foods, our post-hoc hypothesis is that green TL labels may be useful when healthy foods are less palatable. A recent study showed that TL labeling was not advantageous compared to GDA [22], which may be due to subjective perceived enjoyment of the food. Therefore, the effects of TL and GDA nutrition labels on choices involving unpalatable foods should be compared in future research. In addition, society considers healthy food advantageous for physical health. Therefore, there may have been a social desirability effect involved in the selection of healthy food in the experiments. Moreover, the participants orally reported after experiment that they were not aware of the meaning of the color label; hence, their healthy food choices were not influenced by prior knowledge of what the labels signified. In contrast, a difference was found between the two nutrition labels for unhealthy foods, indicating that TL labels indeed have an effect on unhealthy food choice. In future study, the social desirability effect should be controlled to more accurately explore the effect of TL labels on healthy food.

Several limitations of the present study should be noted. Dieting is the most popular method of weight loss [23]; therefore, we specifically recruited dieters to investigate the possible effects of nutritional labels on food selection. However, a previous study showed that TL labels are likely to be a cost-effective method for preventing obesity on a behavioral level [15]. As we only recruited participants who were chronic dieters and of healthy weight, it is unclear if this same effect would be seen in participants who are overweight or obese. Further research using fMRI should be conducted to confirm whether TL nutrition labels are equally effective for obese individuals. Furthermore, the current study lacked a control condition, i.e., making a choice with no label present, and therefore baseline food choice in the absence of labels could not be examined. In addition, there are individual differences in food preferences. Although this study attempted to control for the taste of food, it could not resolve this issue entirely. Therefore, personalized experimental materials should be used in future studies to avoid the influence of individual differences.

Future studies should address the effectiveness of nutrition labels in specific situations. For example, it would be advantageous to study dieters during states of negative emotion or resource depletion because they often display emotional eating tendencies when faced with stress or negative emotions [43] and their food-specific inhibitory control is often decreased after depletion of cognitive resources [41]. Whether appropriate nutrition labels can block unhealthy dietary decisions in these situations is an important subject for further study. In addition, the currently used process of food choice-related decision-making is not ecologically valid. Future research should simulate online food purchases, providing corresponding numerical nutritional information or more prominent nutritional labels alongside the orders. In addition to exploring the underlying neurobiological effects of nutrition labels, it would be beneficial to also track eye movements during simulated online food purchases to better elucidate how nutrition labels alter unhealthy dietary choices from an attentional perspective.

## 5. Conclusions

In sum, the present study found that, compared with GDA label, salient TL labels may reduce unhealthy food choice decisions by increasing the activity in brain regions implicated in response or motor inhibition and by increasing their connectivity with the ACC, a salient information monitoring region. The use of red TL labels can potentially inform dieters and help them stop selecting unhealthy dietary options and may be a more effective public policy intervention than is the use of GDA labels.

## Figures and Tables

**Figure 1 nutrients-12-00128-f001:**
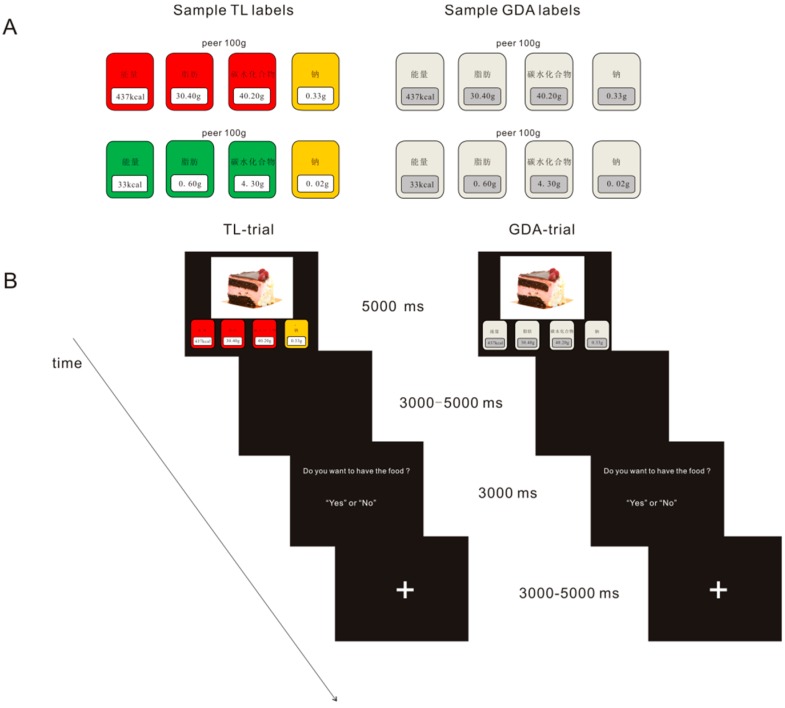
Food label decision making task. (**A**) Sample labels generated for the foods. Left: color-coded traffic-light (TL) label for unhealthy (upper) and healthy (lower) foods. Right: guideline-daily-amount (GDA) label for the same foods. The food nutrition label is presented to the participants in Chinese characters, and their English translations are, from left to right: 能量 = energy, 脂肪 = fat, 碳水化合物 = carbohydrates, and 钠 = sodium. (**B**) Illustration of the trial setup.

**Figure 2 nutrients-12-00128-f002:**
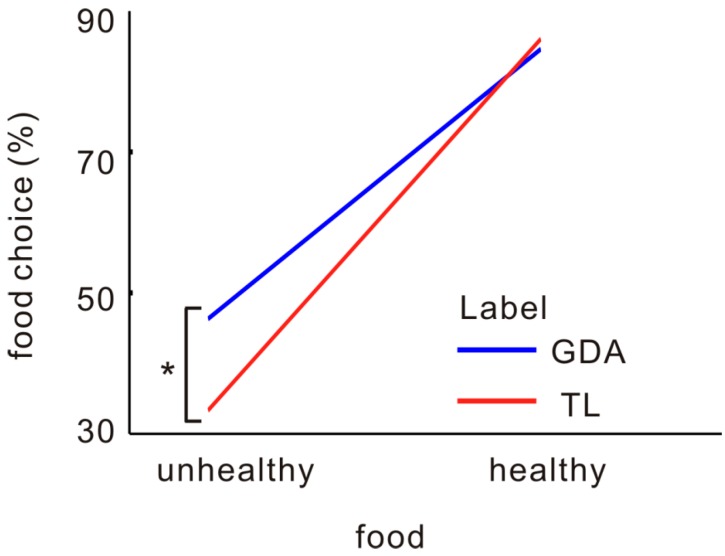
A comparison of food choices between the label conditions. GDA = guideline-daily-amount condition; TL = traffic-light-like condition; * *p* < 0.05.

**Figure 3 nutrients-12-00128-f003:**
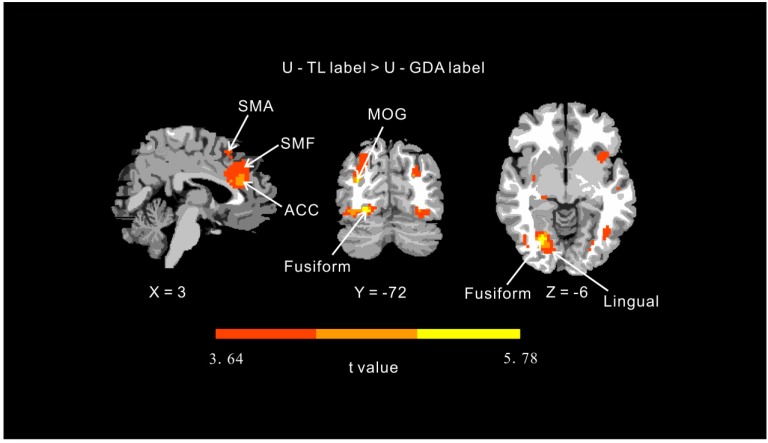
Contrast maps for presentation of unhealthy foods with guideline-daily-amount (GDA) and color-coded traffic-light (TL) labels. Note: U = unhealthy food, SMA = supplementary motor area, SMF = superior medial frontal gyrus, MOG = middle occipital gyrus, ACC = anterior cingulate cortex.

**Figure 4 nutrients-12-00128-f004:**
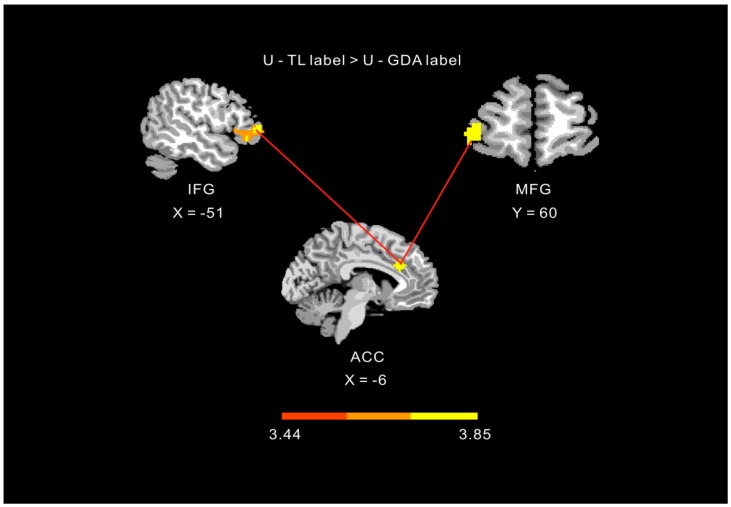
Functional connectivity results for presentation of unhealthy foods with guideline-daily-amount (GDA) and color-coded traffic-light (TL) labels. Note: U = unhealthy food, ACC = anterior cingulate cortex, IFG = inferior frontal gyrus, MFG = middle frontal gyrus.

**Table 1 nutrients-12-00128-t001:** Significant results of the fMRI analyses during unhealthy food presentation with GDA and color-coded TL labels.

Effect	Brain Area	Laterality	Cluster Size	*x*	*y*	*z*	*t*
U-TL label > U-GDA label ^a^	Anterior cingulate cortex	L	98	−6	27	27	4.13
	Medial frontal gyrus	L	65	−3	30	42	3.66
	Supplementary motor area		25	0	18	48	3.87
	Fusiform gyrus	R	112	24	−72	−9	5.78
	Lingual gyrus	R	64	12	−78	0	3.64
	Middle occipital gyrus	R	147	30	−84	17	3.98
PPI: H-TL label > H-GDA label	Inferior frontal gyrus	L	104	−51	45	3	3.85
(seed area: Anterior cingulate cortex) ^b^	Middle frontal gyrus	R	22	39	60	0	3.51
	Middle temporal gyrus	R	213	60	−24	−6	3.44

Note: fMRI = functional magnetic resonance imaging; GDA = guideline daily amount; TL = traffic light; U = unhealthy food; PPI = psychophysiological interaction; ^a^ whole-brain corrected for topological false discovery rate *p* < 0.001; ^b^ whole-brain corrected for topological false discovery rate *p* < 0.005.

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
