# Peer review of "Red for “Stop”: “Traffic-Light” Nutrition Labels Decrease Unhealthy Food Choices by Increasing Activity and Connectivity in the Frontal Lobe"

_nutrients, 2020, doi:10.3390/nu12010128_

Round 1
Reviewer 1 Report
The sentence: "... whether they wanted it 19 or not " in the Abstract is very unformal.Methods:
The rationale for female subjects only is not stated. The rational for this particular sample (chronic dieters) should be stated more detailed: Why do you expect significant chances/results from these subjects? What about other diets (exclusion criteria?) The 2 hour fasting is a bit too short for subjects to induce cravings, you need at least 3 hours before the blood sugar will drop. Please justify. Please explain why the internal consistency for the DEBQ scale is so low in your study. Was age added as a covariate? This is not stated in the paper.Results:
Maybe neural changes were influence by the color of the item (colored vs. grey picture). Have you considered this in your analyses? If so, how?Author Response
Dear editor and the reviewers,
Thank you for your constructive comments on our paper (nutrients-650108). We are very grateful to the editor and reviewers for the excellent level of detailed feedback offered to enable us to enhance the manuscript. We have carefully addressed the comments of the reviewers and highlighted (in red) the main changes made in the revised paper. All responses are made as follows.
Sincerely,
Xuemeng Zhang
Reviewer(s)' Comments to Author:
Comments from the editors and reviewers:
-Reviewer 1
The sentence: "... whether they wanted it 19 or not " in the Abstract is very unformal.
Response: Thank you for your valuable advice for improving our study. We have revised it. And we have hired an English-language editor to revise the writing of paper.
“Forty-four female dieters (age, mean = 20.0, standard deviation = 1.45 years) were recruited to participate in a food-choice task; healthy or unhealthy food options were presented with color-coded traffic-light or purely numeric guideline-daily-amount labels, and the participants were asked to state their preference.” (line 16 to 19)
Methods:
The rationale for female subjects only is not stated. The rational for this particular sample (chronic dieters) should be stated more detailed: Why do you expect significant chances/results from these subjects? What about other diets (exclusion criteria?) The 2 hour fasting is a bit too short for subjects to induce cravings, you need at least 3 hours before the blood sugar will drop. Please justify. Please explain why the internal consistency for the DEBQ scale is so low in your study. Was age added as a covariate? This is not stated in the paper.
Response: Thank you for your constructive advice for improving our study. Because restricted diet is common among women and male dieters are relatively rare (Kong et al., 2011) [39], to avoid imbalance in the number of recruited men and women, only women were selected to participate in this study. We have stated it in the text (line 123 to 125).
“Because restricted diet is common among women and male dieters are relatively rare [39], to avoid imbalance in the number of recruited men and women, only women were selected to participate in this study.”
The 2 hours fasting was refer to previous study (van der Laan, de Ridder, Charbonnier, Viergever, & Smeets, 2014; Zhou, Gao, Chen, & Kong, 2017). But it’s indeed too short for subjects to induce craving. We have revised our wording (line 140 to 142). The 2 hours fasting was for control hunger level rather than for induce cravings.
“Participants were asked to refrain from eating or drinking (except water) for at least 2 h before arriving at the laboratory to control the hunger level”.
In this study, the lower internal consistency coefficient of DEBQ is due to the calculation based on 44 individual data (sample size of the current study). Since the number of subjects in the fMRI empirical study cannot reach the number of questionnaire studies, the internal consistency coefficient is lower than previous studies.
The age was not as a covariate, because we recruit college students to control age, and the standard deviation of age is very small (age, mean [M] = 20.0, standard deviation [SD] = 1.45). We detail this in the methods section (line 125 to 127).
“Moreover, to avoid the effects of age, our study sample was limited to college students (age, mean [M] = 20.0, standard deviation [SD] = 1.45).”
Kong, F. C., Zhang, Y., Chen, H., Shi, M. L., Todd, J., & Gao, X. (2011). The cognitive bias in food cues for restricted eaters: Evidences from behavioral and neuropsychological studies. Advances in Psychological Science, 19(9), 1355-1362.
van der Laan, L. N., de Ridder, D. T., Charbonnier, L., Viergever, M. A., & Smeets, P. A. (2014). Sweet lies: neural, visual, and behavioral measures reveal a lack of self-control conflict during food choice in weight-concerned women. Frontiers in behavioral neuroscience, 8, 184.
Zhou, Y., Gao, X., Chen, H., & Kong, F. (2017). High-disinhibition restrained eaters are disinhibited by self-regulatory depletion in the food-related inhibitory control. Eating behaviors, 26, 70-75.
Results:
Maybe neural changes were influence by the color of the item (colored vs. grey picture). Have you considered this in your analyses? If so, how?
Response: Thank you for your careful review of our manuscript. The aim of present study was to investigate the effect of prominent color nutrition labels on behavior and brain, to prove that labels with warning colors are more effective. Therefore, the neural changes were influenced by the color of the item, which can prove our assumption. And there was different brain activity between colored label and grey label in the results.
Reviewer 2 Report
Major:
Lines 142-144- Why was the stimulus order not randomized? It appears you just had 4 Identical runs.
Lines 230-231 the p-value is sufficiently strict however there is no mention of clustering or similar false discovery rate techniques. My understanding is that topological methods rely on random field theory. It has been argued that imaging data is not sufficiently smooth to meet the criteria for the application of random field and such smoothing would eliminate one of the main advantages of MRI. I suggest that at a minimum clustering is conducted or non-parametric measures are employed (see, “Cluster failure: Why fMRI inferences for spatial extent have inflated false-positive rates”). At a minimum the authors should provide sufficient rational for the use of the current FDR technique.
Lines 247-248 Justification should be given for the use of the 0.005 alpha level. This reads to me as though nothing was significant at the 0.001 level and so the authors chose to alter their initial alpha level. If so this should be detailed out in the text and the 0.001 level should be reported.
Lines 333-335 this is a stretch to say that it is the connectivity between these regions that drove the reduction in selection of the foods.
Lines 338-340 if this null result can be attributed to the materials and population then so can the positive result.
Lines 341-345 Healthy foods are rarely considered palatable or particularly appealing. The fact that 90% of the healthy items were chosen speaks to the bias observed in this population. It is likely that this experiment is running into issues with social desirability. This could also possibly explain the results. For example, the drop in high ed food selection could be simply because the red label indicates that they shouldn’t select it so they don’t because they know the researchers will see their response. Conversely, it is possible that rather than a drop in actual selection we are seeing a rise in selection of unhealthy foods due to the lack of the red label. It is difficult to tease apart directionality because there is no control condition (i.e., making a choice with no label present). This is particularly relevant as all the foods were liked by the participant so one would assume that similar amounts of foods would be selected if this bias did not exist.
Lines 354-363 the authors discuss one major limitation and one minor. However there are several large issues that need to be addressed here.
Minor:
Abstract:
This sentence is very hard to follow: “Forty-four female dieters (age, mean = 20.0, standard deviation = 1.45) were recruited to 18 participate in a food choice task, indicating, for healthy or unhealthy food presented with a color19 coded traffic-light label or a purely numeric guideline-daily-amount label, whether they wanted it 20 or not.”
Lines 219-220 Why were participants discarded completely? Why not discard the run or censor the value?
Lines 324-325 This study has nothing to do with consumption but with choice. The authors should be clear that it may help some individuals to avoid certain high energy products but it is unknown of the products that they still choose if they will overconsume them or not just because of the red symbols
Author Response
Dear editor and the reviewers,
Thank you for your constructive comments on our paper (nutrients-650108). We are very grateful to the editor and reviewers for the excellent level of detailed feedback offered to enable us to enhance the manuscript. We have carefully addressed the comments of the reviewers and highlighted (in red) the main changes made in the revised paper. All responses are made as follows.
Sincerely,
Xuemeng Zhang
Reviewer(s)' Comments to Author:
Comments from the editors and reviewer:
-Reviewer 2
Major:
Lines 142-144- Why was the stimulus order not randomized? It appears you just had 4 Identical runs.
Response: Thank you for your constructive advice for improving our study. We elaborate on the sequence of experimental trials presented in the method section (line 146 to 149). The experimental trials were randomly arranged in advance to ensure that the experimental conditions are not repeated multiple times in succession, so that different experimental conditions would appear intermittently.
“Each run included healthy and unhealthy foods with a TL or GDA label and lasted 398 s. Each run contained 24 trials, including four types of conditions. The experimental trials were randomly arranged in advance to ensure that the experimental conditions were not repeated multiple times in succession, so that different experimental conditions would appear intermittently.”
Lines 230-231 the p-value is sufficiently strict however there is no mention of clustering or similar false discovery rate techniques. My understanding is that topological methods rely on random field theory. It has been argued that imaging data is not sufficiently smooth to meet the criteria for the application of random field and such smoothing would eliminate one of the main advantages of MRI. I suggest that at a minimum clustering is conducted or non-parametric measures are employed (see, “Cluster failure: Why fMRI inferences for spatial extent have inflated false-positive rates”). At a minimum the authors should provide sufficient rational for the use of the current FDR technique.
Response: Thank you for your constructive advice for improving our study. We have provided rational for the use of the current FDR technique in the 2.7.2 section (line 235 to 241).
“The original voxel-wise FDR considers that voxels are independent of each other. However, voxels are not independent of each other after fMRI data smoothing, which causes the FDR obtained using this voxel-wise FDR correction to be very high for an activated brain region. The topological false discovery rate (FDR) is based on the gaussian random field theory, which considers that voxels are not independent. Voxels beyond the T threshold constitute a cluster, and then FDR is used to correct these clusters, so that the independent units (used to be the total number of voxels, but now is the number of clusters) are greatly reduced [46]. Therefore, the statistical threshold was set at a whole-brain corrected value with a topological FDR of p < 0.001 to visualize the main effects.”
And we did find that topological FDR correction was more effective than voxel-wise FDR correction (The following picture on the left used original voxel-wise FDR correction, and the right picture used topological FDR correction).
Lines 247-248 Justification should be given for the use of the 0.005 alpha level. This reads to me as though nothing was significant at the 0.001 level and so the authors chose to alter their initial alpha level. If so this should be detailed out in the text and the 0.001 level should be reported.
Response: Thank you for your constructive advice for improving our study. The significance level was initial set at 0.001 for PPI results, we fail to find brain region, so we set at 0.005 to explore the results. We have explained the significance level setting in the text (line 260 to 262).
“The statistical threshold was initially set at a whole-brain corrected value with a topological FDR of p < 0.001 to visualize main effects. However, as no brain regions were found at the 0.001 level, we further set it to p < 0.005 to explore the results.”
And we found that there were many worthy studies with the same situation, the threshold of activation is set more strictly, while the PPI threshold was set weakly (Diekhof, & Gruber, 2010; Grabenhorst, Schulte, Maderwald, & Brand, 2013; Joassin et al., 2011; Kanat, Heinrichs, Schwarzwald, & Domes, 2015; Peters, & Büchel, 2010; Townsend et al., 2013; Van Overwalle, & Mariën, 2016). Given different analysis methods with different thresholds were accepted in many studies, so we have adopted this way to reported results.
Diekhof, E. K., & Gruber, O. (2010). When desire collides with reason: functional interactions between anteroventral prefrontal cortex and nucleus accumbens underlie the human ability to resist impulsive desires. Journal of Neuroscience, 30(4), 1488-1493.
Grabenhorst, F., Schulte, F. P., Maderwald, S., & Brand, M. (2013). Food labels promote healthy choices by a decision bias in the amygdala. Neuroimage, 74, 152-163.
Joassin, F., Pesenti, M., Maurage, P., Verreckt, E., Bruyer, R., & Campanella, S. (2011). Cross-modal interactions between human faces and voices involved in person recognition. Cortex, 47(3), 367-376.
Kanat, M., Heinrichs, M., Schwarzwald, R., & Domes, G. (2015). Oxytocin attenuates neural reactivity to masked threat cues from the eyes. Neuropsychopharmacology, 40(2), 287.
Peters, J., & Büchel, C. (2010). Episodic future thinking reduces reward delay discounting through an enhancement of prefrontal-mediotemporal interactions. Neuron, 66(1), 138-148.
Townsend, J. D., Torrisi, S. J., Lieberman, M. D., Sugar, C. A., Bookheimer, S. Y., & Altshuler, L. L. (2013). Frontal-amygdala connectivity alterations during emotion downregulation in bipolar I disorder. Biological psychiatry, 73(2), 127-135.
Van Overwalle, F., & Mariën, P. (2016). Functional connectivity between the cerebrum and cerebellum in social cognition: a multi-study analysis. NeuroImage, 124, 248-255.
Lines 333-335 this is a stretch to say that it is the connectivity between these regions that drove the reduction in selection of the foods.
Response: Thank you for your careful review of our manuscript. This is indeed an inferential result. We rephrased this part (line 348 to 350).
“Additionally, the red TL labels promoted functional connectivity between the corresponding food value computations and activity in inhibitory brain regions, possibly contributing to the reduction of unhealthy decisions.”
Lines 338-340 if this null result can be attributed to the materials and population then so can the positive result.
Response: Thank you for your valuable advice for improving our study. In the discussion section (line 362 to 366), we have explained the results of unhealthy food with experimental materials and subjects.
“Moreover, highly palatable but unhealthy foods are not conducive to a dieter's weight control; therefore, such foods were selected to a lesser degree than healthy foods. Especially, dieters reduced their choice of unhealthy foods under the red TL label than under the GDA label condition, suggesting that the red TL nutrition label could effectively activate cognitive resources to prevent the participant from selecting the food.”
Lines 341-345 Healthy foods are rarely considered palatable or particularly appealing. The fact that 90% of the healthy items were chosen speaks to the bias observed in this population. It is likely that this experiment is running into issues with social desirability. This could also possibly explain the results. For example, the drop in high ed food selection could be simply because the red label indicates that they shouldn’t select it so they don’t because they know the researchers will see their response. Conversely, it is possible that rather than a drop in actual selection we are seeing a rise in selection of unhealthy foods due to the lack of the red label. It is difficult to tease apart directionality because there is no control condition (i.e., making a choice with no label present). This is particularly relevant as all the foods were liked by the participant so one would assume that similar amounts of foods would be selected if this bias did not exist.
Response: Thank you for your constructive advice for improving our study. The social desirability was possible an effect on results. We have discussed the possible consequences of social desirability in line 371 to 375. However, the subjects indicated that they did not know the meaning of the color in the post hoc report (we had stated it in the procedure section, line 152 to 154), so we consider that the subjects' reduction of unhealthy food choices was caused by the salience traffic signal labels, and we have discussed this in line 375 to 377. The lack of control conditions is indeed a deficiency of this study, and we have explained in the limitations (line 385 to 387).
“In addition, society considers healthy food advantageous for physical health. Therefore, there may have been a social desirability effect involved in the selection of healthy food in the experiments. Moreover, the participants orally reported after experiment that they were not aware of the meaning of the color label; hence, their healthy food choices were not influenced by prior knowledge of what the labels signified. In contrast, a difference was found between the two nutrition labels for unhealthy foods, indicating that TL labels indeed have an effect on unhealthy food choice. In future study, the social desirability effect should be controlled for to more accurately explore the effect of TL labels on healthy food.”
“Furthermore, the current study lacked a control condition, i.e., making a choice with no label present, and therefore baseline food choice in the absence of labels could not be examined.”
Lines 354-363 the authors discuss one major limitation and one minor. However there are several large issues that need to be addressed here.
Response: Thank you for your valuable advice for improving our study. Based on your suggestion, we have elaborated on the limitations of the lack of control conditions (line 377 to 378) and the influence of social desirability effect (line 385 to 387).
“In future study, the social desirability effect should be controlled for to more accurately explore the effect of TL labels on healthy food.”
“Furthermore, the current study lacked a control condition, i.e., making a choice with no label present, and therefore baseline food choice in the absence of labels could not be examined.”
Minor:
Abstract:
This sentence is very hard to follow: “Forty-four female dieters (age, mean = 20.0, standard deviation = 1.45) were recruited to 18 participate in a food choice task, indicating, for healthy or unhealthy food presented with a color19 coded traffic-light label or a purely numeric guideline-daily-amount label, whether they wanted it 20 or not.”
Response: Thank you for your valuable advice for improving our study. We have revised it. And we have hired an English-language editor to revise the writing of paper.
“Forty-four female dieters (age, mean = 20.0, standard deviation = 1.45 years) were recruited to participate in a food-choice task; healthy or unhealthy food options were presented with color-coded traffic-light or purely numeric guideline-daily-amount labels, and the participants were asked to state their preference.” (line 16 to 19)
Lines 219-220 Why were participants discarded completely? Why not discard the run or censor the value?
Response: Thank you for your valuable advice for improving our study. Because their heads are too large, brain imaging results may be artifacts. These participants discarded completely was due to the insufficient number of effective trials for these people, which would affect the quality of data.
Lines 324-325 This study has nothing to do with consumption but with choice. The authors should be clear that it may help some individuals to avoid certain high energy products but it is unknown of the products that they still choose if they will overconsume them or not just because of the red symbols
Response: Thank you for your constructive advice for improving our study. We have revised our wording (line 339 to 340).
“This provides potential evidence that red TL labels for unhealthy foods convey a negative health message, thereby allowing individuals to refrain from selecting such foods.”
